The role of cerebral blood flow volume in cortical inhibition during postural changes

Sagirov Arlan F. arlansagirov@gmail.com
Sergeev Timofey V.
Kuropatenko Maria V.
Shabrov Alexander V.
Agapova Elizaveta A.
Anisimov Alexey A.
Yafarov Alexander Z.
Department of Ecological Physiology, Federal State Budgetary Scientific Institution “Institute of Experimental Medicine” , Saint-Petersburg , Russia
Silva Nuno
Electronic publication date: 2025 Oct 27
Publication date: 2025
Volume: 13
Electronic Location ID: e20233
Received 2025 Feb 5; Accepted 2025 Sep 23
Copyright: ©2025 Sagirov et al.
Copyright year: 2025
Copyright holder: Sagirov et al.
License: This is an open access article distributed under the terms of the Creative Commons Attribution License, which permits unrestricted use, distribution, reproduction and adaptation in any medium and for any purpose provided that it is properly attributed. For attribution, the original author(s), title, publication source (PeerJ) and either DOI or URL of the article must be cited.
License URL: https://creativecommons.org/licenses/by/4.0/

Keywords: Cortical inhibition, EEG, Alpha waves, Brain bioelectrical activity, Rheoencephalography, REG, Brain blood flow, Postural changes, Baroreflex, Tilt table

Funding: Ministry of Science and Higher Education of Russian Federation This work was supported by the Ministry of Science and Higher Education of the Russian Federation as part of the state assignment “Development of technology for restoring functional reserves and adaptive potential to maintain active and healthy longevity of life”. The funders had no role in study design, data collection and analysis, decision to publish, or preparation of the manuscript.

==============================
This study investigated how body posture impacts cerebral hemodynamics and brain bioelectrical activity, aiming to understand the mechanisms by which increased cerebral blood flow in a supine position might lead to cortical inhibition, potentially indicated by a reduction in alpha wave presence. The study also explored the neurovascular effects of dynamic tilting. Simultaneous electroencephalographic (EEG) and rheoencephalographic (REG) recordings were conducted on 40 healthy participants (mean age = 21.3 ± 1.4 years; 20 men and 20 women) during two postural tests. In Test 1, participants transitioned between sitting upright and lying supine. Test 2 followed a similar design, with the addition of dynamic tilting through passive oscillations between +10° and −10° on a tilt table. Results indicated that REG parameters –specifically rheographic wave amplitude (RWA), venous outflow (VO), and catacrotic time (CT) –increased notably in the supine position, particularly among male participants. In Test 1, men also exhibited a pronounced drop in alpha absolute spectral power (Pα) when moving from upright to supine, while Pα remained relatively stable in women. In Test 2, Pα showed minimal changes among men, whereas moderate reductions were observed in women, mainly in the supine position following dynamic tilting. Significant sex differences were noted in RWA, VO, and Pα, with these values generally higher in female participants. A strong negative correlation between RWA and Pα was observed in the male group during Test 1, with a similar but weaker trend in women. In Test 2, a negative correlation between RWA and Pα re-emerged in both groups, though it did not reach statistical significance. These findings suggest that baroreflex activity may be the primary driver of cortical inhibition, with changes in cerebral blood flow volume potentially playing a secondary role. Dynamic tilting had minimal impact on brain blood flow and mixed effects on Pα, though results may hint at a possible interference with baroreflex responses, which could attenuate cortical inhibition. Overall, this study demonstrates the use of combined REG and EEG during postural transitions as a tool for investigating the interaction between cerebral blood flow and brain activity. These findings and the methods used may have clinical relevance as a potential diagnostic approach for disorders involving impaired baroreflex function.

Introduction

Body position significantly impacts the functional states of the cardiovascular, central, and autonomic nervous systems. Postural effects are typically studied using various tests, such as orthostatic and clinostatic tests, where individuals actively change their posture, and the tilt test, which uses a specialized tilt table to position the subject. These methods are commonly employed in medical and physical evaluations of patients with orthostatic disorders (Aponte-Becerra & Novak, 2021; Cheshire et al., 2021). In some cases, these assessments are supplemented with instrumental techniques, including Doppler ultrasonography, magnetic resonance imaging (MRI), electroencephalographic (EEG), and others, to provide a more comprehensive evaluation. Such combinations have been extensively used to investigate the effects of microgravity on the human body, particularly its influence on brain bioelectrical activity, cerebral blood flow, and cerebrospinal fluid circulation.

Electrophysiological research has shown that a supine or slightly antiorthostatic position dampens brain alpha waves compared to an upright stance (Vaitl et al., 1996; Schneider et al., 2008; Spironelli, Busenello & Angrilli, 2016; Spironelli & Angrilli, 2017). Studies by Thibault, Lifshitz & Raz (2016); Chang et al. (2011) demonstrated that higher frequencies in the brain’s EEG spectrum increase as the body adopts a more vertical position. Body posture has been found to modulate perception and cognitive processing by lowering pain thresholds and slowing cognitive and emotional responses (Spironelli & Angrilli, 2011; Fardo, Spironelli & Angrilli, 2013; Messerotti Benvenuti, Bianchin & Angrilli, 2013; Jung, Cho & Kang, 2020). Many studies emphasize the dynamic relationship between brain bioelectrical activity and cerebral hemodynamics during postural changes. Physiologically, the effects of posture on brain activity and hemodynamics may be attributed to the stimulation of carotid baroreceptors in a supine position. This stimulation reduces sympathetic activation and lowers noradrenergic output from the locus coeruleus, ultimately leading to cortical inhibition—a phenomenon observed as a decrease in alpha wave presence (Dworkin et al., 1994; Rau & Elbert, 2001; Berridge & Waterhouse, 2003; Lipnicki, 2009). These baroreceptors are sensitive to blood pressure changes, which are accompanied by blood and cerebrospinal fluid (CSF) volume redistribution in the intracranial space during postural transitions, a phenomenon useful in clinical assessments of postural hemodynamics (Goldstein & Cheshire Jr, 2017; Baker et al., 2024).

In our study, we simultaneously recorded brain bioelectrical activity, focusing on changes in the alpha wave spectrum, alongside cerebral hemodynamics during two postural tests using EEG and rheoencephalography (REG). EEG is a well-established electrophysiological method for detecting spontaneous brain bioelectrical patterns or evoked potentials via scalp electrodes. EEG signals primarily arise from synchronized post-synaptic potentials within neuronal assemblies linked to information processing (Buzsaki, Anastassiou & Koch, 2012; Schomer & Lopes da Silva, 2017). The alpha rhythm, in the 8–16 Hz range, is a widely studied frequency band and is easily detectable in the occipital and parietal regions of awake individuals with closed eyes. Alpha waves may also appear in other cortical areas and coexist with delta, theta, beta, and gamma bands.

REG is a non-invasive method for assessing cerebral blood flow by measuring bioelectrical impedance, which changes in proportion to pulsatile blood volume within the brain (Jenkner, 1968; Yarullin, 1983; Bodo, 2010; Bodó, 2020). In clinical and experimental settings for over a century, REG has been employed in diagnosing intracranial hypertension, vascular conditions, brain trauma, migraines, and for monitoring treatment outcomes. REG recordings are influenced by systemic arterial pressure, heart rate fluctuations, central venous pressure, and intracranial pressure. The REG waveform reflects pulsatile blood flow changes that are dependent on the tone of cerebral vascular walls, allowing REG to capture structural changes (such as those in atherosclerosis) and dynamic tone responses to functional demands. REG captures approximately 60% of blood flow from the hemisphere directly beneath the electrodes, while contributions from the opposite hemisphere and extracranial vessels account for about 25% and 10%, respectively. Thus, REG provides a reasonably accurate measure of intracranial hemodynamics. However, interpretation challenges and variability in rheographic waveform parameters have led to some dissatisfaction among clinicians and physiologists (Montgomery & Gleason, 1992). REG’s application in studies of postural influence on cerebral blood flow remains limited, and the simultaneous use of EEG and REG in this context has been sporadically reported in the literature (Thuróczy et al., 1994; Bodo et al., 2018; Meghdadi et al., 2019).

In this study, we aimed to evaluate the effectiveness of the combined EEG-REG approach for investigating brain blood flow and alpha activity across various postures and during passive oscillatory movements on a tilt table (referred to here as dynamic tilting). Given the existing knowledge on intracranial hemodynamics and the baroreceptor reflex, our primary objective was to explore the underlying mechanisms by which increased blood and CSF volume in the brain during the supine position, may lead to cortical inhibition, defined in this study as a decrease in alpha wave spectral power (Pα). Additionally, we sought to investigate the potential neurovascular effects of dynamic tilting.

Materials and Methods

Participants

Forty participants (mean age = 21.32 ± 1.37 years; 20 men and 20 women) provided written informed consent, following the guidelines of the Institute of Experimental Medicine’s Ethics Committee (No. 2/22, dated 06.04.2022) and in compliance with the Declaration of Helsinki. All participants reported normal or corrected-to-normal vision and no history of serious health issues related to cardiovascular disease, neurological disorders, or traumatic injuries. Female participants were admitted to the study within seven days before or after their supposed ovulation date. Twenty-nine participants (16 men and 13 women) completed the study by participating in both tests.

Procedure

Before the trials, participants received a brief overview of the upcoming tests. Throughout the EEG-REG recordings, they were instructed to relax and remain still with their eyes closed. The study consisted of two tests involving simultaneous EEG and REG recordings. Each test was conducted on a separate day.

The first test included three EEG-REG recordings: first, participants were instructed to sit in an armchair in a relaxed upright position for 3 min; next, they proceeded to lie supine on a tilt table set horizontally for 8 min; finally, they were told to return to a relaxed seated position in the armchair for an additional 3 min.

The second test followed a similar structure, but introduced dynamic tilting. Participants followed the instructions by sitting still in the armchair for 3 min, then lay in a supine horizontal position on the tilt table for 2 min. Afterward, participants were alerted that the tilt table will initiate a swinging motion, and they should remain relaxed. The tilt table was oscillating between +10° and −10° at a rate of one full cycle per minute (from 0° to −10°, then to +10°, and back to 0°). After completing four oscillation cycles, the tilt table was returned to the horizontal position where participants were asked to lie for an additional 2 min. Finally, they were instructed to transition to a second upright sitting position in the armchair and remain there motionless for 3 min (Fig. 1).

Figure 1 A schematic representation of the two test protocols.

The blue and red wave icons within the boxes represent simultaneous EEG and REG recordings. The smooth blocks above each box indicate the duration of recording in each position. Dotted blocks within the boxes show the specific samples (e.g., SA1, SA2, oHA1) selected for further analysis. * For the 8-minute horizontal supine position in the first test, two average values were calculated by dividing the recording into two 4-minute segments, with two samples taken from the first 4 min and two from the final 4 min. This setup allowed for comparison with the two shorter horizontal positions in the second test. SA and oSA–first sitting upright positions in Test 1 and Test 2; H–horizontal supine position in Test 1; HA–first 2 min of horizontal supine position in Test 1; HB–last 2 min of horizontal supine position in Test 1; oHA–first horizontal supine position before dynamic tilting in Test 2; oHB–second horizontal supine position after dynamic tilting in Test 2; SB and oSB–second sitting upright positions in Test 1 and Test 2.

Simultaneous EEG and REG recordings were conducted in each position, except during the passive oscillatory movements on the tilt table. During each position, blood pressure and heart rate were also measured alongside the EEG and REG recordings.

Equipment

A multichannel topographic electroencephalograph (“Mitsar-EEG”) with 21 active gel electrodes was used to record bioelectrical activity from participants’ scalps in various postures. Cerebral hemodynamic parameters were recorded using a rheographic device (“Diamant-R”), which can perform REG measurements with six electrodes. Two electrodes were placed symmetrically on the forehead, approximately two cm below the frontal EEG electrodes (Fp1 and Fp2), two were positioned on the mastoids, and two near the occipital protuberance. These electrodes formed four channels: left and right fronto-mastoid (FM) and occipito-mastoid (OM) leads, which indicate blood flow changes in four major cerebral arteries—the left and right internal carotid and vertebral arteries. To properly capture REG waveforms, the rheograph requires simultaneous ECG recordings, achieved by placing two additional gel electrodes on participants’ wrists. The six REG electrodes on the head were secured with a specialized rubber band, and a mesh helmet was placed on top to attach the 21 EEG electrodes according to the “10-20” system (Fig. 2). Passive oscillatory movements were applied using an automated tilt table, with distal fixation of the upper and lower extremities. The direction and speed of the tilt table’s motions were controlled via specialized computer software. Blood pressure and heart rate were measured with an automatic blood pressure monitor (OMRON M2 Classic).

Figure 2 Combined arrangement of EEG and REG electrodes.

REG electrodes are shown as red circles, while EEG electrodes, positioned according to the “10-20” system, are represented by blue and white circles. Data from the electrodes marked with white circles was excluded from the subsequent analysis.

Data processing

EEG data was recorded and analyzed using “WinEEG” version 2.130.101 (Ponomaryov, 2018). Electroencephalograms were captured with monopolar leads, referencing an averaged ear electrode. Recording speed was set to 30 mm/s, with a high-pass filter at 0.16 Hz, a low-pass filter at 30 Hz, and a notch filter in the 45–55 Hz range. To minimize artifacts, native EEGs were first visually inspected, followed by an automatic artifact detection and removal process based on the following criteria: wave amplitude of 100 µV or higher, slow waves with an amplitude of 50 µV and a frequency up to 1 Hz, and fast waves with an amplitude of 50 µV and a frequency between 20 and 35 Hz. Following artifact correction, EEG data underwent spectral analysis to calculate alpha wave absolute spectral power, which will be referred for simplicity as alpha spectral power further in the text. Data from electrodes Fp1, Fp2, Fpz, O1, O2, and Oz were excluded from further analysis due to proximity to REG electrodes, which often caused interference and signal distortion.

REG data was recorded at a speed of 25 mm/s and processed using the “Diamant” software package, version 11.06 (Volkov et al., 2018). Due to the robustness of the REG signal, recordings were easily scanned visually, and any distorted waveforms were excluded from analysis. The following parameters were included in the quantitative analysis of REG data: rheographic wave amplitude (RWA) in Ohms, venous outflow (VO) as a percentage, and anacrotic (AT) and catacrotic time (CT) in seconds. RWA indicates the peak intensity of volumetric blood flow in the studied brain area, typically ranging from 0.05 to 0.25 Ohms (Yarullin, 1983; Zenkov & Ronkin, 2014). VO, a derivative of RWA, is a primary indicator of cerebral venous hemodynamics, usually varying between 0% and 25%. AT represents the blood inflow period until peak intensity, indicating the tone and elasticity of cerebral arteries, with an average normal AT of 0.1 s in younger individuals, though it can range from 0.09 to 0.2 s. CT, associated with the descending part of the REG wave, reflects the tone of cerebral veins, typically ranging from 0.2 to 0.7 s.

To calculate alpha spectral power and REG parameters, samples of 20 to 30 s were taken from the EEG and REG recordings. For both tests, two samples were taken from each position, except for the 8-minute horizontal position, from which four samples were extracted for computational analysis. This approach yielded eight samples of EEG and REG recordings for each participant per test. Values for alpha spectral power and REG parameters from two samples in a given position were averaged to represent that position. For the 8-minute horizontal position in the first test, recordings were divided into two 4-minute segments, with two samples taken from the first 4 min and two from the final 4 min.

Statistical analysis

Statistical analysis was conducted using the “Prism 10” software package. The average values of Pα and REG parameters for each position were tested for normality using four tests: Anderson-Darling, D’Agostino & Pearson, Shapiro-Wilk, and Kolmogorov-Smirnov. A dataset was considered normally distributed if it passed at least two of these tests. Based on the normality results, differences between body positions, as well as pre- and post-passive oscillatory movement comparisons, were assessed using one-way repeated measures ANOVA with Tukey’s multiple comparison test for normally distributed data across all body positions, or the nonparametric Friedman test with Dunn’s multiple comparison test if values in any position did not meet normality criteria. All p-values were adjusted for multiple comparisons.

A priori power analysis was conducted to determine which of the observed parameter changes were meaningful within the scope of this study. Given that the total sample size was predetermined and limited to 36–40 participants—with 17–20 participants per sex group—we calculated the required effect size needed to consider meaningful differences for each sample. Subsequently, we estimated the achieved effect size for each sample, assuming a normal distribution of parameter values.

All calculations were performed using G*Power 3.1.9.7 (Faul et al., 2007), with input parameters set according to the relevant statistical tests used under the assumption of normality. These tests were primarily repeated measures ANOVA or ordinary one-way ANOVA. The significance level (α) was set at 0.05, and statistical power (1–β) at 0.8. We assumed a correlation among repeated measures of 0.5 and a nonsphericity correction of 1.

For the total sample (36–40 participants), the required effect size was calculated to be 0.2. For analyses stratified by sex (n = 17–20 per group), the required effect size increased to 0.3. In the case of the one-way ANOVA used to assess sex differences in α spectral power and REG parameters, the total number of values distributed across four groups ranged from 64 to 74, corresponding to a required minimum effect size of 0.43.

The achieved effect sizes were estimated in G*Power using η2, which was derived by dividing the between-group or treatment sum of squares by the total sum of squares. Full results are presented in Tables S1–S3.

For correlation analysis between Pα and RWA values from each sample were averaged and paired according to their respective EEG electrode and REG lead. For instance, Pα values from electrodes F3, F7, C3, P3, and T5 were paired with RWA values from the left FM or OM leads. As the data did not sufficiently meet the criteria for a Gaussian distribution, the Spearman correlation test was used. Simple linear regression was also performed to illustrate sequential changes in Pα and RWA.

Results

Our primary aim was to investigate the impact of posture on REG parameters and alpha spectral power by comparing upright sitting and lying supine positions, as well as to determine whether dynamic tilting influences these electrophysiological characteristics.

Postural hemodynamic changes

Changes in blood pressure and heart rate among participants showed no significant differences between the two tests (see Tables S4 and S5). Both male and female participants experienced a slight but noticeable drop in systolic arterial pressure (SAP) of approximately four mm Hg by the end of the horizontal position in Test 1, with similar changes observed primarily in male participants during Test 2. Diastolic arterial pressure (DAP) and heart rate both decreased substantially during the supine horizontal positions in both tests, with DAP showing an 8–13% reduction and heart rate a 9–16% reduction compared to the upright sitting position. Dynamic tilting had no noticeable effect on these cardiovascular parameters in subsequent body positions.

Postural changes in REG parameters

The most significant changes in REG parameters were observed in the male group compared to females. However, overall, both male and female participants demonstrated a significant effect of posture on RWA, VO, and CT. Among male participants in Test 1, RWA increased by approximately 16% in the left FM (Friedman statistic = 15.06, p = 0.0018) and 13% in the right FM (F (1.490, 28.31) = 7.594, p = 0.0047) when in the supine position compared to the sitting positions (Fig. 3, Fig. S2).

In Test 2, similar RWA increases were observed among male participants, with RWAoHA (M = 0.1438 Ohm, SD = 0.0378) and RWAoHB (M = 0.13 Ohm, SD = 0.035) significantly higher than RWAoSA (M = 0.1256 Ohm, SD = 0.0371) and RWAoSB (M = 0.1172 Ohm, SD = 0.038) in the left FM, with comparable results in the right FM (Fig. S5). In contrast, the female group exhibited only minor RWA fluctuations in both tests, which were not substantial (Figs. S3, S6). When considering overall changes in RWA across all participants, regardless of sex, Test 1 showed a slight decrease only in the second sitting position in the right FM. In Test 2, RWA increased by up to 10% and 13% in the left and right FM, respectively, during supine positions (Figs. S5, S7). A marked difference in RWA between male and female participants was observed, with this disparity being more pronounced in Test 2 (Figs. S8–S11). On average, RWA in the female group was 19% higher in Test 1 and 30% higher in Test 2 compared to males.

Figure 3 Postural changes in RWA among male participants during Test 1.

The graphs illustrate RWA data from the left and right fronto-mastoid (FM) leads in sitting and supine positions. Confidence intervals are displayed alongside the data points, with means represented by circles and medians by diamonds. Colors differentiate between the first sitting (SA) and the initial 4 min of the supine (HA) position, as well as the second sitting (SB) and the final 4 min of the supine (HB) position. “*” –p < 0.05, “**” –p < 0.01.

The RWA alterations were accompanied by corresponding changes in VO, which were more pronounced in the male group. In Test 1, VO increased in the supine position (Figs. S12, S13). Significant VO increases occurred in the left FM among males (F (1.526, 29) = 9.645, p = 0.0014), with VOHA (M = 12.47%, SD = 5.61) and VOHB (M = 13.37%, SD = 5.31) exceeding VOSA (M = 8.66%, SD = 4.25) and VOSB (M = 9.29%, SD = 4.03). A similar trend was observed in the left FM among females (F (1.824, 29.19) = 6.621, p = 0.0052), where VOHA (M = 13.58%, SD = 6.6) and VOHB (M = 14.05%, SD = 5.7) were also higher than VOSA (M = 9.11%, SD = 4.9) and VOSB (M = 10.62%, SD = 4.18). In general, the total sample showed similar results (Fig. S14). In Test 2, the male group showed substantial VO increases in the supine position across three leads (right and left FM, left OM), while the female group exhibited only mild VO increases in the left REG leads (Figs. S15, S16). The total sample demonstrated 28–35% increase in VO in the left REG leads (Fig. S17). Sex differences in VO were more evident in Test 2 (Figs. S18–S20), with VO in females approximately 37% higher than in males, particularly in the upright position.

AT showed minimal variation, despite some statistically significant differences between sitting and supine positions in one REG lead in both tests (Figs. S21–S24). Due to the small effect size and limited variability, AT appeared to remain stable overall.

In contrast, CT displayed a considerable increase in the supine position compared to sitting upright. The most substantial and consistent CT increases were observed in male participants, with an average increase of 19% in Test 1 and 21% in Test 2 (Fig. 4, Figs. S25, S28). The female group demonstrated an overall 11% increase in CT in Test 1 and an 8% increase in Test 2 (Figs. S26, S29). Consequently, same increments in CT were observed in the total sample as well (Figs. S27, S30).

Figure 4 Postural changes in CT among male participants during Test 1.

The graphs display RWA data from the left and right fronto-mastoid (FM) leads in both sitting and supine positions, with confidence intervals shown alongside the data points. Mean values are represented by circular points. Colors differentiate between the initial sitting (SA) and first 4 min of the supine (HA) position, as well as the second sitting (SB) and final 4 min of the supine (HB) position. A one-way repeated measures ANOVA summary for statistically significant results: left FM (F (1.259, 23.91) = 61.4, p < 0.0001), right FM (F (1.384, 26.29) = 34.39, p < 0.0001). “***” –p < 0.001, “****” –p < 0.0001.

In summary, we found that REG parameters—RWA, VO, and CT—increased during the supine horizontal position, with these changes being more pronounced in the male participants.

Postural changes in alpha spectral power

Visual assessment of participants’ EEGs in the upright sitting position revealed that brain bioelectrical activity was primarily characterized by alpha wave spindles with a frequency of 9–11 Hz and an average amplitude of around 60 µV. These alpha waves were most prominent in the occipital regions, with reduced presence in the central regions. Occasionally, the alpha waves were interrupted by low-frequency beta activity of similar amplitude, and sometimes by brief flashes of 5–8 Hz theta waves in the frontal and central regions. When participants shifted to the supine position, a noticeable reduction in native EEG alpha activity, or “alpha depression,” was observed. Subsequent quantitative analysis of Pα confirmed this decrease during the supine position.

The most substantial and significant decrease in Pα occurred in Test 1 among male participants. In the fronto-temporal region of the scalp, recorded by electrodes Fz, F3, F4, F7, and F8, the average reduction in Pα during the supine position was approximately 43% (Figs. S31, S40). Meanwhile, Pα in the central (C3, C4, Cz) and temporal (T4, T3, T5, and T6) regions decreased by 48% and 35%, respectively (Figs. S34, S37, S40). The most pronounced decline in Pα occurred in the parietal region (P3, P4, Pz), with an approximate reduction of 52% (Fig. 5, Figs. S37, S40).

Figure 5 Postural changes in alpha spectral power for the P3 and P4 electrodes in male participants during Test 1.

The graphs present confidence intervals, with median values represented by diamond-shaped points. Colors differentiate between the initial sitting (SA) and first 4 min of the supine (HA) position, as well as the second sitting (SB) and final 4 min of the supine (HB) position. A nonparametric Friedman test summaries for statistically significant results: P3 (Friedman statistic = 24.79, p < 0.0001), P4 (Friedman statistic = 40.71, p < 0.0001). “*” –p < 0.05, “**” –p < 0.01, “***” –p < 0.001, “****” –p < 0.0001.

Conversely, postural changes had a less pronounced effect on the female group in Test 1 (Figs. S32, S35, S38, S41). Statistically significant reductions in Pα of approximately 13% were observed in the supine position at electrodes C4 (Friedman statistic = 13.13, p = 0.0044), T3 (Friedman statistic = 10.58, p = 0.0143), and T4 (Friedman statistic = 10.95, p = 0.012).

In Test 1, the overall sample tended to follow changes in Pα that resembled those observed in the male participants (Figs. S33, S36, S39, S42).

In Test 2, a reduction in Pα during the supine position was more evident among female participants, though not as marked as the changes seen in the male group in Test 1. On average, the female group showed a 10% decrease in Pα in the fronto-temporal region (Figs. S44, S53) and a 15% reduction in the central scalp region (Figs. S47, S53). In the parietal region, an average decrease of 25% in Pα was observed, with statistically significant reductions at electrode P4 (Friedman statistic = 15.56, p = 0.0014) (Figs. S50, S53).

In the male group during Test 2, Pα was less affected by the supine position than in Test 1. A significant decrease of approximately 20% was detected only at the C3 electrode (F (2.075, 31.13) = 5.457, p = 0.0087) (Figs. S43, S46, S49, S52).

The total sample exhibited a 21–27% decrease in Pα across the recorded regions, primarily during the second supine position (Figs. S45, S48, S51, S54).

Additionally, in Test 1, the difference in Pα between sexes was not significant in the upright sitting position; however, in the supine position, the average Pα in the female group was notably higher, exceeding that of the male group by 58% (Figs. S55, S56). Test 2 also showed a tendency for female participants to maintain a higher Pα than their male counterparts, with an approximate 46% difference in both upright and supine positions (Figs. S55–S59).

Overall, both tests demonstrated that Pα levels differed by sex, with the female group consistently displaying higher Pα values than the male group. In Test 1, Pα in females remained relatively stable, with only minor decreases in select areas in the supine position. In contrast, male participants experienced a pronounced drop in Pα when transitioning from an upright to a supine position in Test 1. In Test 2, the male group showed minimal changes in Pα, whereas the female group exhibited moderate reductions, primarily in the supine position following dynamic tilting.

Correlation and regression analysis of RWA and Pα

In the first test, the male group exhibited a strong, significant negative correlation between RWA and Pα across most regions recorded by REG and EEG, except the ones recorded by the left OM (Table 1). Simple linear regression analysis further supported these findings (Fig. 6).

Table 1 Correlation results between RWA and Pα in Test 1 (n = 8).

		F3	F7	C3	P3	T5	
Pα/ RWA	Males	
LFM	r	−0.8333	−0.8333	−0.8333	−0.8095	−0.8333	
p	0.0154	0.0154	0.0154	0.0218	0.0154	
LOM	r	−0.2755	−0.2755	−0.2755	−0.3234	−0.2755	
p	0.5076	0.5076	0.5076	0.4329	0.5076	
		F4	F8	C4	P4	T6	
RFM	r	−0.6826	−0.7785	−0.7785	−0.7785	−0.7785	
p	0.0729	0.0301	0.0301	0.0301	0.0301	
ROM	r	−0.8264	−0.7545	−0.7545	−0.7545	−0.7545	
p	0.0163	0.0377	0.0377	0.0377	0.0377	
Pα/ RWA	Females	
LFM	r	−0.2156	0.1198	−0.0958	−0.0479	0.1916	
p	0.61	0.7797	0.8251	0.9177	0.6479	
LOM	r	−0.241	−0.0482	0.0482	0.0843	0.1687	
p	0.5651	0.9218	0.9218	0.8464	0.6935	
		F4	F8	C4	P4	T6	
RFM	r	−0.4762	−0.4048	−0.5714	−0.7857	−0.2857	
p	0.2431	0.3268	0.1511	0.0279	0.5	
ROM	r	−0.1190	−0.0952	0.4048	0.2381	−0.1667	
P	0.7930	0.8401	0.3268	0.5821	0.7033	
Notes.

LFM left FM

RFM right FM

LOM left OM

ROM right OM

r Spearman’s correlation coefficient

p statistical significance

Statistically significant results are highlighted by bold text. The value of n equals to the number of pairs of averaged RWA and Pα between participants in each analyzed sample of EEG and REG (SA1, SA2, HA1, etc).

Figure 6 Simple linear regression graphs showing estimated relationship between RWA and Pα in all 4 REG leads and several corresponding EEG electrodes among male participants in Test 1.

Each point of certain color represents estimated relationship between RWA and Pα in the corresponding pair of REG lead and EEG electrode, for example, left FM and F3 (LFM-F3), right OM and C4 (RFM-C4), etc. The lines indicate the general trend of the relationship. “*” –p < 0.05, “**” –p < 0.01, “***” –p < 0.001, “****” –p < 0.0001.

Overall, the following patterns were observed:

• In the region recorded by the left FM and corresponding left EEG electrodes (F3, F7, P3, C3, and T5), each 0.01 Ohm increase in RWA was associated with a 1.69 µV2 decrease in Pα.

• In the region recorded by the right FM and right EEG electrodes (F4, F8, P4, C4, and T6), each 0.01 Ohm increase in RWA corresponded to a 1.75 µV2 decrease in Pα.

• In the right OM region with the same right-side EEG electrodes, each 0.01 Ohm increase in RWA was linked to a 3.3 µV2 decrease in Pα.

These results suggest a pronounced inverse relationship between RWA and Pα in the male group, particularly in the regions covered by the FM leads.

In Test 1, female participants showed a strong, significant negative correlation between RWA and Pα, but only in regions recorded by the right FM in conjunction with P4 electrodes. However, further regression analysis did not reveal any statistically significant trends (Fig. S60).

In Test 2, both male and, to a lesser degree, female participants exhibited weak to moderate negative correlations between RWA and Pα, primarily in areas recorded by both FMs and corresponding EEG electrodes (Table S6). Among the female group, only one statistically significant correlation was found: between RWA recorded at the left OM and Pα at C3 (r =  − 0.7545, p = 0.038). Additionally, some significant positive correlations were observed in male participants around the right OM and electrodes F3, C3, and P3. Simple linear regression analysis did not yield any substantial or consistent results for either group in Test 2 (Figs. S62, S63).

An assessment of the correlation between RWA and Pα in the total sample revealed a greater number of significant negative correlations in Test 1 compared to Test 2 (Table S7). Accordingly, the results of the simple linear regression analysis reflected a similar trend (Figs. S61, S64).

In summary:

• A strong negative correlation between RWA and Pα was clearly observed in the male group during Test 1, with a similar, albeit weaker, trend in the female group.

• In Test 2, a negative correlation between RWA and Pα reappeared in both groups, though it was generally not statistically significant.

• Regression analysis highlighted a definitive inverse relationship between RWA and Pα in the total sample, and specifically in the male group during Test 1, supporting the pronounced effect of posture on these parameters in the participants.

Discussion

This study provides evidence on how body posture influences brain bioelectric activity and cerebral hemodynamics, and it explores the relationship between these two factors.

Overall, the observed changes in RWA and VO during postural tests suggest increased cerebral blood volume when transitioning from an upright to a horizontal position. This increase likely elevates intracranial blood pressure for a moment, and activates a baroreflex mechanism that induces vasodilation and reduces cardiac output. The drops DAP and heart rate in the supine position further support these cardiovascular adjustments. It also appears that the increase in cerebral blood volume primarily affects the brain’s venous system when supine, as indicated by notable and consistent increases in VO and CT relative to RWA. This differential increase in venous volume between upright and supine positions may be due to changes in venous drainage pathways (Zenkov & Ronkin, 2014; Valdueza et al., 2000; Gehlen, Kurtcuoglu & Schmid Daners, 2017). Specifically, in an upright position, venous blood flow relies heavily on the vertebral and extrajugular veins, which remain patent within the neck tissues. During supine rest, however, the internal jugular veins become the primary drainage route, supporting a flow rate that increases to approximately 720 ml/min, with only a minor flow of 46 ml/min remaining in the vertebral plexus (Simka, Czaja & Kowalczyk, 2019; Schreiber et al., 2003).

Additionally, observed sex differences in RWA and VO suggest baseline cerebrovascular variations between males and females. Female participants demonstrated higher cerebral blood volume, particularly in the upright position, which may be attributed to the vasodilatory effects of estrogen on the cerebrovascular system (Gisolf et al., 2004). This could also reflect heightened activation of neurohumoral systems in men, specifically the sympathoadrenal system, which contributes to lower pulse blood supply to the brain. Such sex-based cerebrovascular differences emerge during puberty and are generally absent in childhood (Nakajima et al., 1995; Poskotinova & Kamenchenko, 2011). This baseline increase in cerebral blood volume among women may limit their range for further volumetric fluctuations in response to postural shifts, resulting in smaller RWA and VO changes when transitioning from sitting to supine positions.

AT remained stable across most conditions, showing only minor, statistically significant increases in the left or right OM lead during supine positions, though these were inconsistent and insufficient for interpretation. Comparing REG parameters, blood pressure, and heart rate between the two tests showed negligible differences, suggesting that mild dynamic tilting between +10° and −10° does not significantly impact cerebrovascular hemodynamics.

The observed decrease in Pα in the supine position—particularly prominent among male participants in Test 1 and moderately evident among females in Test 2—may indicate cortical inhibition. These changes could be attributed to baroreflex reactions from carotid baroreceptors, with postural blood redistribution potentially playing a secondary or coincidental role. This hypothesis is further supported by correlation and regression analysis showing an inverse relationship between alpha spectral power and peak cerebral blood flow in cortical regions supplied by the internal carotid arteries. The absence of a direct impact from blood redistribution on α-activity is indicated by the generally higher RWA, VO, and Pα values in females than in males. If blood redistribution had a direct effect on α-activity, we would expect female participants, with their higher RWA, to exhibit lower Pα than males, contrary to what was observed. However, despite these findings, the lack of direct comparisons between female and male participants limits the statistical evidence, preventing definitive conclusions about the observed sex differences in REG parameters and alpha spectral power.

Dynamic tilting had mixed effects on Pα when comparing Test 1 and Test 2. While it appeared to dampen Pα reductions during the second supine position in males, it increased the likelihood of Pα decreases in females during the same period. In Test 2, where dynamic tilting was applied, the correlation between RWA and Pα was generally weak and inconsistent. This discrepancy may suggest differences in hormonal states, anxiety levels, subtle variations in neurovascular responses between men and women, and the interplay of baroreflex and cerebral blood flow fluctuations induced by tilting, which may obscure the relationship between α-activity and cerebral blood flow. The lack of assessment of participants’ state anxiety prior to testing may be considered a study limitation, along with the absence of measurements such as norepinephrine levels, pupil diameter, dynamic oxygen tension, and heart rate variability—factors that could have further supported our hypothesis.

Conclusion

This study demonstrates that brain blood flow volume increases in the supine position, primarily within the cerebral venous system, while a decrease in alpha spectral power, which is one of the indicators of cortical inhibition, is also observed in this position. Usually, cortical inhibition appears to stem mainly from baroreflex activity originating in the carotid sinus, which, through thalamic and locus coeruleus pathways, reduces cortical activity. The observed differences in RWA and VO between male and female participants may be attributed to higher estrogen levels in women, leading to vasodilation and vessel proliferation, and to the sympathoadrenal activity in men, which limits pulse blood supply to the brain. Notably, the consistently higher Pα values in females compared to males suggest that increased cerebral blood flow does not inherently lead to reduced α-activity. This may strengthen the suggestion that the baroreflex mechanism can contribute to the process of cortical inhibition to some extent.

Dynamic tilting, as applied in the second test, had a variable effect on alpha spectral power and REG parameters, likely creating short-term, localized hemodynamic fluctuations. It is unclear whether these fluctuations may or may not interfere with baroreflex responses, reducing their influence on cortical α-activity.

In considering the clinical significance of these findings, the combined use of EEG, REG, and postural tests—with potential refinements—holds promise as a diagnostic tool for conditions involving impaired sympathetic baroreflex function, such as Parkinson’s disease, multiple system atrophy, and dementia with Lewy bodies (Goldstein, 2003; Goldstein & Sharabi, 2019; Chen, Li & Liu, 2020; Goldstein & Sharabi, 2023).

Supplemental Information

Supplemental Information 1 Achieved effect size (fa) for total sample in both tests

nnd –non-Gaussian distribution, ns –insignificant results. The table demonstrates achieved effect size for each parameter that had significant changes in both tests. Values of achieved effect size (fa) that are equal or exceed the value of correspondent required effect size (fr) calculated for every sample are highlighted by green color.

Supplemental Information 2 Achieved effect size (fa) for male and female samples in both tests

nnd –non-Gaussian distribution, ns –insignificant results. The table demonstrates achieved effect size for each parameter that had significant changes in both tests. Values of achieved effect size (fa) that are equal or exceed the value of correspondent required effect size (fr) calculated for every sample are highlighted by green color.

Supplemental Information 3 Achieved effect size (fa) for the sample used in assessment of sex differences in α spectral power and REG parameters

nnd –non-Gaussian distribution, ns –insignificant results. The table demonstrates achieved effect size for each parameter that had significant changes in both tests. Values of achieved effect size (fa) that are equal or exceed the value of correspondent required effect size (fr) calculated for every sample are highlighted by green color.

Supplemental Information 4 SAP, DAP and heart rate changes during different body position in Test 1

All samples were normally distributed. M and SD are mean value and standard deviation. Asterisk (*) points out the values in horizontal supine position that significantly differed from sitting upright (p < 0.05). The underlined values indicate statistical difference between them in two horizontal supine positions or within the same horizontal supine position (p < 0,05).

Supplemental Information 5 SAP, DAP and heart rate changes during different body position in Test 2

All samples were normally distributed. M and SD are mean value and standard deviation. Asterisk (*) points out the values in horizontal supine position that significantly differed from sitting upright (p < 0.05). The underlined values indicate statistical difference between them in two horizontal supine positions or within the same horizontal supine position (p < 0.05).

Supplemental Information 6 Correlation results between RWA and Pα in Test 2 (n= 8)

LFM –left FM, RFM –right FM, LOM –left OM, ROM –right OM. r –Spearman’s correlation coefficient, p –statistical significance. Statistically significant results are highlighted by green color. The value of n equals to the number of pairs of averaged RWA and Pα between participants in each analyzed sample of EEG and REG (oSA1, oSA2, oHA1, etc.).

Supplemental Information 7 Correlation results between RWA and Pα among all participants in both tests (n= 8)

LFM –left FM, RFM –right FM, LOM –left OM, ROM –right OM. r –Spearman’s correlation coefficient, p –statistical significance. Statistically significant results are highlighted by green color. The value of n equals to the number of pairs of averaged RWA and Pα between participants in each analyzed sample of EEG and REG (oSA1, oSA2, oHA1, etc.).

Supplemental Information 8 The studied parameters of REG wave

Horizontal dashed line represents isoline, and vertical dashed lines are the parameters explored in this study. Av and A3/4are main components in estimation of venous output (VO). Anacrotic (AT) and catacrotic (CT) time are also shown as arrowed lines. RWA –reowave amplitude, Av –amplitude of the maximum systolic value of REG wave venous component, A3/4 –reowave amplitude in the last quarter of cardiac cycle.

Supplemental Information 9 Postural changes of RWA among male participants during Test 1 (n = 20)

The graphs show data from 4 REG leads: left and right fronto-mastoid (FM), left and right occcipito-mastoid (OM) for sitting and supine positions. The graphs show confidence intervals with means represented by circle-shaped points, and medians depicted as rhomb-shaped points. Additionally, points and intervals are highlighted by different colors to distinguish between first sitting (SA) and first 2 min of supine (HA) position and second sitting (SB) and last 2 min of supine (HB) position. A one-way repeated measures ANOVA and a nonparametric Friedman test summaries for statistically significant results: left FM (Friedman statistic = 15.05, p = 0.0018), right FM (F (1.490, 28.31) = 7.594, p = 0.0047). “*” –p < 0.05, “**” –p < 0.01.

Supplemental Information 10 Postural changes of RWA among female participants during Test 1 (n = 17)

The graphs show data from 4 REG leads: left and right fronto-mastoid (FM), left and right occcipito-mastoid (OM) for sitting and supine positions. The graphs show confidence intervals with means represented by circle-shaped points, and medians depicted as rhomb-shaped points. Additionally, points and intervals are highlighted by different colors to distinguish between first sitting (SA) and first 2 min of supine (HA) position and second sitting (SB) and last 2 min of supine (HB) position.

Supplemental Information 11 Postural changes of RWA among all participants during Test 1 (n = 37)

The graphs show data from 4 REG leads: left and right fronto-mastoid (FM), left and right occcipito-mastoid (OM) for sitting and supine positions. The graphs show confidence intervals with means represented by circle-shaped points, and medians depicted as rhomb-shaped points. Additionally, points and intervals are highlighted by different colors to distinguish between first sitting (SA) and first 2 min of supine (HA) position and second sitting (SB) and last 2 min of supine (HB) position. A one-way repeated measures ANOVA summary for statistically significant results: right FM (F (1.665, 59.93) = 7.849, p = 0.0018). “*” –p < 0.05, “**” –p < 0.01.

Supplemental Information 12 Postural changes of RWA among male participants during Test 2 (n = 16)

The graphs show data from 4 REG leads: left and right fronto-mastoid (FM), left and right occcipito-mastoid (OM) for sitting and supine positions. The graphs show confidence intervals with means represented by circle-shaped points, and medians depicted as rhomb-shaped points. Additionally, points and intervals are highlighted by different colors to distinguish between first sitting (oSA) and supine (oHA) positions and second sitting (oSB) and supine (oHB) positions. A one-way repeated measures ANOVA summary for statistically significant results: left FM (F (1.348, 20.23) = 5.213, p = 0.0247), right FM (F (1.583, 23.74) = 12.83, p = 0.0004). “*” –p < 0.05, “**” –p < 0.01, “***” –p < 0.001.

Supplemental Information 13 Postural changes of RWA among female participants during Test 2 (n = 16)

The graphs show data from 4 REG leads: left and right fronto-mastoid (FM), left and right occcipito-mastoid (OM) for sitting and supine positions. The graphs show confidence intervals with means represented by circle-shaped points, and medians depicted as rhomb-shaped points. Additionally, points and intervals are highlighted by different colors to distinguish between first sitting (oSA) and supine (oHA) positions and second sitting (oSB) and supine (oHB) positions.

Supplemental Information 14 Postural changes of RWA among all participants during Test 2 (n = 32)

The graphs show data from 4 REG leads: left and right fronto-mastoid (FM), left and right occcipito-mastoid (OM) for sitting and supine positions. The graphs show confidence intervals with means represented by circle-shaped points, and medians depicted as rhomb-shaped points. Additionally, points and intervals are highlighted by different colors to distinguish between first sitting (oSA) and supine (oHA) positions and second sitting (oSB) and supine (oHB) positions. A one-way repeated measures ANOVA summary for statistically significant results: left FM (F (1.949, 60.42) = 8.408, p = 0.0007), right FM (F (2.101, 60.92) = 15.55, p < 0.0001). “*” –p < 0.05, “***” –p < 0.001, “****” –p < 0.0001.

Supplemental Information 15 Sex differences in RWA during sitting positions in Test 1 (n= 37)

The graphs show data from 4 REG leads: left and right fronto-mastoid (FM), left and right occcipito-mastoid (OM) for two sitting positions (SA and SB). Black boxplots include values of male participants (m), and red boxplots contain values of female participants (f). Pairs of boxplots were analyzed separately using one-way ANOVA, i.e., SA (m) was compared only to SA (f), and SB (m) was compared only to SB (f). Outliers are shown by black and blue points. A one-way ANOVA and a nonparametric Kruskal–Wallis test summaries for statistically significant results: left FM (Kruskal–Wallis statistic = 14.54, p = 0.0023), right FM (F (3, 70) = 7.007, p = 0.0003). “**” –p < 0.01.

Supplemental Information 16 Sex differences in RWA during supine positions in Test 1 (n= 37)

The graphs show data from 4 REG leads: left and right fronto-mastoid (FM), left and right occcipito-mastoid (OM) for the first and last 2 min of supine position (HA and HB). Black boxplots include values of male participants (m), and red boxplots contain values of female participants (f). Pairs of boxplots were analyzed separately using one-way ANOVA, i.e., HA (m) was compared only to HA (f), and HB (m) was compared only to HB (f). Outliers are shown by black and blue points. A one-way ANOVA test summary for statistically significant results: right FM (F (3, 70) = 3.595, p = 0.0177). “*” –p < 0.05.

Supplemental Information 17 Sex differences in RWA during sitting positions in Test 2 (n= 32)

The graphs show data from 4 REG leads: left and right fronto-mastoid (FM), left and right occcipito-mastoid (OM) for two sitting positions (oSA and oSB). Black boxplots include values of male participants (m), and red boxplots contain values of female participants (f). Pairs of boxplots were analyzed separately using one-way ANOVA, i.e., oSA (m) was compared only to oSA (f), and oSB (m) was compared only to oSB (f). A one-way ANOVA and a nonparametric Kruskal–Wallis test summaries for statistically significant results: left FM (F (3, 60) = 6.97, p = 0.0004), right FM (F (3, 60) = 8.796, p < 0.0001), left OM (F (3, 60) = 3.276, p < 0.027), right OM (Kruskal–Wallis statistic = 9.816, p = 0.0202). “*” –p < 0.05, “**” –p < 0.01

Supplemental Information 18 Sex differences in RWA during supine positions in Test 2 (n= 32)

The graphs show data from 4 REG leads: left and right fronto-mastoid (FM), left and right occcipito-mastoid (OM) for two supine positions (oHA and oHB). Black boxplots include values of male participants (m), and red boxplots contain values of female participants (f). Pairs of boxplots were analyzed separately using one-way ANOVA, i.e., oHA (m) was compared only to oHA (f), and oHB (m) was compared only to oHB (f). A one-way ANOVA summary for statistically significant results: left FM (F (3, 60) = 9.697, p < 0.0001), right FM (F (3, 60) = 5.838, p = 0.0014), left OM (F (3, 60) = 9.511, p < 0.0001), right OM (F (3, 60) = 5.145, p = 0.0031). “*” –p < 0.05, “**” –p < 0.01, “***” –p < 0.001.

Supplemental Information 19 Postural changes of VO among male participants during Test 1 (n = 20)

The graphs show data from 4 REG leads: left and right fronto-mastoid (FM), left and right occcipito-mastoid (OM) for sitting and supine positions. The graphs show confidence intervals with means represented by circle-shaped points, and medians depicted as rhomb-shaped points. Additionally, points and intervals are highlighted by different colors to distinguish between first sitting (SA) and first 2 min of supine (HA) position and second sitting (SB) and last 2 min of supine (HB) position. A one-way repeated measures ANOVA summary for statistically significant results: left FM (F (1.526, 29.00) = 9.645, p = 0.0014). “*” –p < 0.05, “**” –p < 0.01.

Supplemental Information 20 Postural changes of VO among female participants during Test 1 (n = 17)

The graphs show data from 4 REG leads: left and right fronto-mastoid (FM), left and right occcipito-mastoid (OM) for sitting and supine positions. The graphs show confidence intervals with means represented by circle-shaped points, and medians depicted as rhomb-shaped points. Additionally, points and intervals are highlighted by different colors to distinguish between first sitting (SA) and first 2 min of supine (HA) position and second sitting (SB) and last 2 min of supine (HB) position. A one-way repeated measures ANOVA summary for statistically significant results: left FM (F (1.824, 29.19) = 6.621, p = 0.0052). “*” –p < 0.05, “**” –p < 0, 01.

Supplemental Information 21 Postural changes of VO among all participants during Test 1 (n = 37)

The graphs show data from 4 REG leads: left and right fronto-mastoid (FM), left and right occcipito-mastoid (OM) for sitting and supine positions. The graphs show confidence intervals with means represented by circle-shaped points, and medians depicted as rhomb-shaped points. Additionally, points and intervals are highlighted by different colors to distinguish between first sitting (SA) and first 2 min of supine (HA) position and second sitting (SB) and last 2 min of supine (HB) position. A one-way repeated measures ANOVA summary for statistically significant results: left FM (F (1.756, 63.23) = 16.34, p < 0.0001). “*” –p < 0.05, “**” –p < 0, 01, “***” –p < 0.001, “****” –p < 0.0001.

Supplemental Information 22 Postural changes of VO among male participants during Test 2 (n = 16)

The graphs show data from 4 REG leads: left and right fronto-mastoid (FM), left and right occcipito-mastoid (OM) for sitting and supine positions. The graphs show confidence intervals with means represented by circle-shaped points, and medians depicted as rhomb-shaped points. Additionally, points and intervals are highlighted by different colors to distinguish between first sitting (oSA) and supine (oHA) positions and second sitting (oSB) and supine (oHB) positions. A one-way repeated measures ANOVA summary for statistically significant results: left FM (F (1.697, 25.46) = 16.03, p < 0.0001), right FM (F (3, 45) = 4.812, p = 0.0054), left OM (F (2.073, 31.10) = 9.858, p = 0.0004). “*” –p < 0.05, “**” –p < 0.01, “***” –p < 0.001.

Supplemental Information 23 Postural changes of VO among female participants during Test 2 (n = 16)

The graphs show data from 4 REG leads: left and right fronto-mastoid (FM), left and right occcipito-mastoid (OM) for sitting and supine positions. The graphs show confidence intervals with means represented by circle-shaped points, and medians depicted as rhomb-shaped points. Additionally, points and intervals are highlighted by different colors to distinguish between first sitting (oSA) and supine (oHA) positions and second sitting (oSB) and supine (oHB) positions.

Supplemental Information 24 Postural changes of VO among all participants during Test 2 (n = 32)

The graphs show data from 4 REG leads: left and right fronto-mastoid (FM), left and right occcipito-mastoid (OM) for sitting and supine positions. The graphs show confidence intervals with means represented by circle-shaped points, and medians depicted as rhomb-shaped points. Additionally, points and intervals are highlighted by different colors to distinguish between first sitting (oSA) and supine (oHA) positions and second sitting (oSB) and supine (oHB) positions. A one-way repeated measures ANOVA and a nonparametric Friedman test summaries for statistically significant results: left FM (F (1.565, 48.53) = 10.37, p = 0.0005), left OM (Friedman statistic = 14.36, p = 0.0025). “*” –p < 0.05, “**” –p < 0.01, “***” –p < 0.001.

Supplemental Information 25 Sex differences in VO during sitting positions in Test 1 (n= 37)

The graphs show data from 4 REG leads: left and right fronto-mastoid (FM), left and right occcipito-mastoid (OM) for two sitting positions (SA and SB). Black boxplots include values of male participants (m), and red boxplots contain values of female participants (f). Pairs of boxplots were analyzed separately using one-way ANOVA, i.e., SA (m) was compared only to SA (f), and SB (m) was compared only to SB (f). Outliers are shown by black and blue points.

Supplemental Information 26 Sex differences in VO during supine positions in Test 1 (n= 37)

The graphs show data from 4 REG leads: left and right fronto-mastoid (FM), left and right occcipito-mastoid (OM) for the first and last 2 min of supine position (HA and HB). Black boxplots include values of male participants (m), and red boxplots contain values of female participants (f). Pairs of boxplots were analyzed separately using one-way ANOVA, i.e., HA (m) was compared only to HA (f), and HB (m) was compared only to HB (f). Outliers are shown by black and blue points.

Supplemental Information 27 Sex differences in VO during sitting positions in Test 2 (n= 32)

The graphs show data from 4 REG leads: left and right fronto-mastoid (FM), left and right occcipito-mastoid (OM) for two sitting positions (oSA and oSB). Black boxplots include values of male participants (m), and red boxplots contain values of female participants (f). Pairs of boxplots were analyzed separately using one-way ANOVA, i.e., oSA (m) was compared only to oSA (f), and oSB (m) was compared only to oSB (f). A one-way ANOVA and a nonparametric Kruskal–Wallis test summaries for statistically significant results: left FM (F (3, 60) = 4.447, p = 0.0069), left OM (F (3, 60) = 3.268, p < 0.0273), right OM (F (3, 60) = 3.794, p = 0.0147). “*” –p < 0.05, “**” –p < 0,01.

Supplemental Information 28 Postural changes of AT among male participants during Test 1 (n = 20)

The graphs show data from 4 REG leads: left and right fronto-mastoid (FM), left and right occcipito-mastoid (OM) for sitting and supine positions. The graphs show confidence intervals with medians depicted as rhomb-shaped points. Additionally, points and intervals are highlighted by different colors to distinguish between first sitting (SA) and first 2 min of supine (HA) position and second sitting (SB) and last 2 min of supine (HB) position.

Supplemental Information 29 Postural changes of AT among female participants during Test 1 (n = 17)

The graphs show data from 4 REG leads: left and right fronto-mastoid (FM), left and right occcipito-mastoid (OM) for sitting and supine positions. The graphs show confidence intervals with medians depicted as rhomb-shaped points. Additionally, points and intervals are highlighted by different colors to distinguish between first sitting (SA) and first 2 min of supine (HA) position and second sitting (SB) and last 2 min of supine (HB) position. A nonparametric Friedman test summary for statistically significant results: right OM (Friedman statistic = 20.64, p = 0.0001). “*” –p < 0.05, “***” –p < 0.001.

Supplemental Information 30 Postural changes of AT among male participants during Test 2 (n = 16)

The graphs show data from 4 REG leads: left and right fronto-mastoid (FM), left and right occcipito-mastoid (OM) for sitting and supine positions. The graphs show confidence intervals with medians depicted as rhomb-shaped points. Additionally, points and intervals are highlighted by different colors to distinguish between first sitting (oSA) and supine (oHA) positions and second sitting (oSB) and supine (oHB) positions. A nonparametric Friedman test summary for statistically significant results: right OM (Friedman statistic = 17.31, p = 0.0006). “*” –p < 0.05.

Supplemental Information 31 Postural changes of AT among female participants during Test 2 (n = 16)

The graphs show data from 4 REG leads: left and right fronto-mastoid (FM), left and right occcipito-mastoid (OM) for sitting and supine positions. The graphs show confidence intervals with medians depicted as rhomb-shaped points. Additionally, points and intervals are highlighted by different colors to distinguish between first sitting (oSA) and supine (oHA) positions and second sitting (oSB) and supine (oHB) positions. A nonparametric Friedman test summary for statistically significant results: left OM (Friedman statistic = 22.05, p < 0.0001). “*” –p < 0.05, “***” –p < 0.001.

Supplemental Information 32 Postural changes of CT among male participants during Test 1 (n = 20)

The graphs show data from 4 REG leads: left and right fronto-mastoid (FM), left and right occcipito-mastoid (OM) for sitting and supine positions. The graphs show confidence intervals with means depicted as circle-shaped points. Additionally, points and intervals are highlighted by different colors to distinguish between first sitting (SA) and first 2 min of supine (HA) position and second sitting (SB) and last 2 min of supine (HB) position. A one-way repeated measures ANOVA summary for statistically significant results: left FM (F (1.259, 23.91) = 61.4, p < 0.0001), right FM (F (1.384, 26.29) = 34.39, p < 0.0001), left OM (F (1.63, 30.98) = 40.61, p < 0.0001), right OM (F (1.636, 31.09) = 19.23, p < 0.0001). “*” –p < 0.05, “***” –p < 0.001, “****” –p < 0.0001.

Supplemental Information 33 Postural changes of CT among female participants during Test 1 (n = 17)

The graphs show data from 4 REG leads: left and right fronto-mastoid (FM), left and right occcipito-mastoid (OM) for sitting and supine positions. The graphs show confidence intervals with means represented by circle-shaped points, and medians depicted as rhomb-shaped points. Additionally, points and intervals are highlighted by different colors to distinguish between first sitting (SA) and first 2 min of supine (HA) position and second sitting (SB) and last 2 min of supine (HB) position. A nonparametric Friedman test summary for statistically significant results: left FM (Friedman statistic = 41.13, p < 0.0001), right FM (Friedman statistic = 41.61, p < 0.0001). “***” –p < 0.001, “****” –p < 0.0001.

Supplemental Information 34 Postural changes of CT among all participants during Test 1 (n = 37)

The graphs show data from 4 REG leads: left and right fronto-mastoid (FM), left and right occcipito-mastoid (OM) for sitting and supine positions. The graphs show confidence intervals with means depicted as circle-shaped points. Additionally, points and intervals are highlighted by different colors to distinguish between first sitting (SA) and first 2 min of supine (HA) position and second sitting (SB) and last 2 min of supine (HB) position. A one-way repeated measures ANOVA summary for statistically significant results: left FM (F (1.404, 47.73) = 89.62, p < 0.0001), right FM (F (1.479, 51.77) = 55.95, p < 0.0001), left OM (F (1.544, 52.49) = 33.78, p < 0.0001), right OM (F (1.850, 66.59) = 14.33, p < 0.0001). “*” –p < 0.05, “***” –p < 0.001, “****” –p < 0.0001.

Supplemental Information 35 Postural changes of CT among male participants during Test 2 (n = 16)

The graphs show data from 4 REG leads: left and right fronto-mastoid (FM), left and right occcipito-mastoid (OM) for sitting and supine positions. The graphs show confidence intervals with means represented by circle-shaped points. Additionally, points and intervals are highlighted by different colors to distinguish between first sitting (oSA) and supine (oHA) positions and second sitting (oSB) and supine (oHB) positions. A one-way repeated measures ANOVA summary for statistically significant results: left FM (F (1.634, 24.51) = 62.31, p < 0.0001), right FM (F (1.582, 23.73) = 54.73, p < 0.0001), left OM (F (1,.804, 27.06) = 33.33, p < 0.0001), right OM (F (1.446, 21.69) = 36.55, p < 0.0001). “*” –p < 0.05, “***” –p < 0.001, “****” –p < 0.0001.

Supplemental Information 36 Postural changes of CT among female participants during Test 2 (n = 16)

The graphs show data from 4 REG leads: left and right fronto-mastoid (FM), left and right occcipito-mastoid (OM) for sitting and supine positions. The graphs show confidence intervals with means represented by circle-shaped points, and medians depicted as rhomb-shaped points. Additionally, points and intervals are highlighted by different colors to distinguish between first sitting (oSA) and supine (oHA) positions and second sitting (oSB) and supine (oHB) positions. A one-way repeated measures ANOVA and a nonparametric Friedman test summaries for statistically significant results : left FM (Friedman statistic = 27.95, p < 0.0001), right FM (F (1.969, 29.54) = 4.443, p = 0.021). “*” –p < 0.05, “**” –p < 0.01, “***” –p < 0.001.

Supplemental Information 37 Postural changes of CT among all participants during Test 2 (n = 32)

The graphs show data from 4 REG leads: left and right fronto-mastoid (FM), left and right occcipito-mastoid (OM) for sitting and supine positions. The graphs show confidence intervals with means represented by circle-shaped points. Additionally, points and intervals are highlighted by different colors to distinguish between first sitting (oSA) and supine (oHA) positions and second sitting (oSB) and supine (oHB) positions. A one-way repeated measures ANOVA summary for statistically significant results: left FM (F (1.665, 49.94) = 67.29, p < 0.0001), right FM (F (1.553, 46.59) = 28.32, p < 0.0001), left OM (F (1.658, 49.74) = 33.86, p < 0.0001), right OM (F (1.165, 33.77) = 4.148, p = 0.044). “***” –p < 0.001, “****” –p < 0.0001.

Supplemental Information 38 Postural changes of alpha spectral power (Pα) calculated for F3, F4, F7 and F8 electrodes among male participants during Test 1 (n = 19)

The graphs show confidence intervals with means represented by circle-shaped points, and medians depicted as rhomb-shaped points. Additionally, points and intervals are highlighted by different colors to distinguish between first sitting (SA) and first 2 min of supine (HA) position and second sitting (SB) and last 2 min of supine (HB) position. A one-way repeated measures ANOVA and a nonparametric Friedman test summaries for statistically significant results: F3 (F (1.840, 33.12) = 13.37, p < 0.0001), F4 (F (1.606, 28.92) = 9.69, p = 0.0012), F7 (Friedman statistic = 26.81, p < 0.0001), F8 (Friedman statistic = 26.59, p < 0.0001). “*” –p < 0.05, “**” –p < 0.01, “****” –p < 0.0001.

Supplemental Information 39 Postural changes of alpha spectral power (Pα) calculated for F3, F4, F7 and F8 electrodes among female participants during Test 1 (n = 16)

The graphs show confidence intervals with means represented by circle-shaped points. Additionally, points and intervals are highlighted by different colors to distinguish between first sitting (SA) and first 2 min of supine (HA) position and second sitting (SB) and last 2 min of supine (HB) position.

Supplemental Information 40 Postural changes of alpha spectral power (Pα) calculated for F3, F4, F7 and F8 electrodes among all participants during Test 1 (n = 33)

The graphs show confidence intervals with means represented by circle-shaped points, and medians depicted as rhomb-shaped points. Additionally, points and intervals are highlighted by different colors to distinguish between first sitting (SA) and first 2 min of supine (HA) position and second sitting (SB) and last 2 min of supine (HB) position. A one-way repeated measures ANOVA and a nonparametric Friedman test summaries for statistically significant results: F3 (F (2.057, 63.76) = 14.65, p < 0.0001), F4 (F (2.016, 62.49) = 11.43, p < 0.0001), F7 (F (2.11, 63.31) = 10.47, p < 0.0001), F8 (Friedman statistic = 28.39, p < 0.0001). “*” –p < 0.05, “**” –p < 0.01, “***” –p < 0.001, “****” –p < 0.0001.

Supplemental Information 41 Postural changes of alpha spectral power (Pα) calculated for C3, C4, T3 and T4 electrodes among male participants during Test 1 (n = 19)

The graphs show confidence intervals with means represented by circle-shaped points, and medians depicted as rhomb-shaped points. Additionally, points and intervals are highlighted by different colors to distinguish between first sitting (SA) and first 2 min of supine (HA) position and second sitting (SB) and last 2 min of supine (HB) position. A one-way repeated measures ANOVA and a nonparametric Friedman test summaries for statistically significant results: C3 (F (1.985, 35.74) = 16.47, p < 0.0001), C4 (Friedman statistic = 21.51, p < 0.0001), T3 (Friedman statistic = 33, p < 0.0001), T4 (Friedman statistic = 28.96, p < 0.0001). “*” –p < 0.05, “**” –p < 0.01, “***” –p < 0.001, “****” –p < 0.0001.

Supplemental Information 42 Postural changes of alpha spectral power calculated for C3, C4, T3 and T4 electrodes among female participants during Test 1 (n = 16)

The graphs show confidence intervals with medians represented by rhomb-shaped points. Additionally, points and intervals are highlighted by different colors to distinguish between first sitting (SA) and first 2 min of supine (HA) position and second sitting (SB) and last 2 min of supine (HB) position. A nonparametric Friedman test summary for statistically significant results: C4 (Friedman statistic = 13.13, p = 0.0044), T3 (Friedman statistic = 10.58, p = 0.0143), T4 (Friedman statistic = 10.95, p = 0.012). “*” –p < 0.05, “**” –p < 0.01.

Supplemental Information 43 Postural changes of alpha spectral power (Pα) calculated for C3, C4, T3 and T4 electrodes among all participants during Test 1 (n = 35)

The graphs show confidence intervals with means represented by circle-shaped points, and medians depicted as rhomb-shaped points. Additionally, points and intervals are highlighted by different colors to distinguish between first sitting (SA) and first 2 min of supine (HA) position and second sitting (SB) and last 2 min of supine (HB) position. A one-way repeated measures ANOVA and a nonparametric Friedman test summaries for statistically significant results: C3 (Friedman statistic = 30.38, p < 0.0001), C4 (F (2.196, 65.89) = 13.24, p < 0.0001), T3 (F (2.083, 64.57) = 9.932, p = 0.0001), T4 (F (2.453, 76.04) = 13.9, p < 0.0001). “*” –p < 0.05, “**” –p < 0.01, “***” –p < 0.001, “****” –p < 0.0001.

Supplemental Information 44 Postural changes of alpha spectral power (Pα) calculated for P3, P4, T5 and T6 electrodes among male participants during Test 1 (n = 19)

The graphs show confidence intervals with means represented by circle-shaped points, and medians depicted as rhomb-shaped points. Additionally, points and intervals are highlighted by different colors to distinguish between first sitting (SA) and first 2 min of supine (HA) position and second sitting (SB) and last 2 min of supine (HB) position. A nonparametric Friedman test summaries for statistically significant results: P3 (Friedman statistic = 24.79, p < 0.0001), P4 (Friedman statistic = 40.71, p < 0.0001), T5 (Friedman statistic = 16.43, p = 0.0009), T6 (Friedman statistic = 28.2, p < 0.0001). “*” –p < 0.05, “**” –p < 0.01, “***” –p < 0.001, “****” –p < 0.0001.

Supplemental Information 45 Postural changes of alpha spectral power (Pα) calculated for P3, P4, T5 and T6 electrodes among female participants during Test 1 (n = 16)

The graphs show confidence intervals with means represented by circle-shaped points. Additionally, points and intervals are highlighted by different colors to distinguish between first sitting (SA) and first 2 min of supine (HA) position and second sitting (SB) and last 2 min of supine (HB) position.

Supplemental Information 46 Postural changes of alpha spectral power (Pα) calculated for P3, P4, T5 and T6 electrodes among all participants during Test 1 (n = 35)

The graphs show confidence intervals with means represented by circle-shaped points, and medians depicted as rhomb-shaped points. Additionally, points and intervals are highlighted by different colors to distinguish between first sitting (SA) and first 2 min of supine (HA) position and second sitting (SB) and last 2 min of supine (HB) position. A nonparametric Friedman test summaries for statistically significant results: P3 (Friedman statistic = 32.54, p < 0.0001), P4 (Friedman statistic = 44.89, p < 0.0001), T5 (Friedman statistic = 19.57, p = 0.0002), T6 (Friedman statistic = 20.77, p = 0.0001). “*” –p < 0.05, “**” –p < 0.01, “***” –p < 0.001, “****” –p < 0.0001.

Supplemental Information 47 Postural changes of alpha spectral power (Pα) calculated for Fz, Cz and Pz electrodes among male participants during Test 1 (n = 19)

The graphs show confidence intervals with means represented by circle-shaped points, and medians depicted as rhomb-shaped points. Additionally, points and intervals are highlighted by different colors to distinguish between first sitting (SA) and first 2 min of supine (HA) position and second sitting (SB) and last 2 min of supine (HB) position. A one-way repeated measures ANOVA and a nonparametric Friedman test summaries for statistically significant results: Fz (F (1.864, 33.55) = 10.67, p = 0.0003), Cz (F (2.353, 42.35) = 11.67, p < 0.0001), Pz (Friedman statistic = 21.76, p < 0.0001). “*” –p < 0.05, “**” –p < 0.01, “***” –p < 0.001.

Supplemental Information 48 Postural changes of alpha spectral power (Pα) calculated for Fz, Cz and Pz electrodes among female participants during Test 1 (n = 16)

The graphs show confidence intervals with means represented by circle-shaped points, and medians depicted as rhomb-shaped points. Additionally, points and intervals are highlighted by different colors to distinguish between first sitting (SA) and first 2 min of supine (HA) position and second sitting (SB) and last 2 min of supine (HB) position.

Supplemental Information 49 Postural changes of alpha spectral power (Pα) calculated for Fz, Cz and Pz electrodes among all participants during Test 1 (n = 35)

The graphs show confidence intervals with means represented by circle-shaped points, and medians depicted as rhomb-shaped points. Additionally, points and intervals are highlighted by different colors to distinguish between first sitting (SA) and first 2 min of supine (HA) position and second sitting (SB) and last 2 min of supine (HB) position. A one-way repeated measures ANOVA and a nonparametric Friedman test summaries for statistically significant results: Fz (Friedman statistic = 32.16, p < 0.0001), Cz (Friedman statistic = 25.25, p < 0.0001), Pz (Friedman statistic = 32.78, p < 0.0001). “*” –p < 0.05, “**” –p < 0.01, “***” –p < 0.001, “****” –p < 0.0001.

Supplemental Information 50 Postural changes of alpha spectral power (Pα) calculated for F3, F4, F7 and F8 electrodes among male participants during Test 2 (n = 16)

The graphs show confidence intervals with means represented by circle-shaped points, and medians depicted as rhomb-shaped points. Additionally, points and intervals are highlighted by different colors to distinguish between first sitting (oSA) and supine (oHA) positions and second sitting (oSB) and supine (oHB) positions.

Supplemental Information 51 Postural changes of alpha spectral power (Pα) calculated for F3, F4, F7 and F8 electrodes among female participants during Test 2 (n = 17)

The graphs show confidence intervals with means represented by circle-shaped points, and medians depicted as rhomb-shaped points. Additionally, points and intervals are highlighted by different colors to distinguish between first sitting (oSA) and supine (oHA) positions and second sitting (oSB) and supine (oHB) positions. A one-way repeated measures ANOVA and a nonparametric Friedman test summaries for statistically significant results: F3 (F (2.125, 34.00) = 3.569, p = 0.0367), F7 (Friedman statistic = 9.918, p = 0.0193), F8 (Friedman statistic = 9.494, p < 0.0234). “*” –p < 0.05.

Supplemental Information 52 Postural changes of alpha spectral power (Pα) calculated for F3, F4, F7 and F8 electrodes among all participants during Test 2 (n = 30)

The graphs show confidence intervals with means represented by circle-shaped points, and medians depicted as rhomb-shaped points. Additionally, points and intervals are highlighted by different colors to distinguish between first sitting (oSA) and supine (oHA) positions and second sitting (oSB) and supine (oHB) positions. A one-way repeated measures ANOVA and a nonparametric Friedman test summaries for statistically significant results: F3 (F (2.302, 69.05) = 5.448, p = 0.0044), F7 (F (2.336, 72.4) = 4.329, p = 0.0126). “*” –p < 0.05, “**” –p < 0.01.

Supplemental Information 53 Postural changes of alpha spectral power (Pα) calculated for C3, C4, T3 and T4 electrodes among male participants during Test 2 (n = 16)

The graphs show confidence intervals with means represented by circle-shaped points, and medians depicted as rhomb-shaped points. Additionally, points and intervals are highlighted by different colors to distinguish between first sitting (oSA) and supine (oHA) positions and second sitting (oSB) and supine (oHB) positions. A one-way repeated measures ANOVA and a nonparametric Friedman test summaries for statistically significant results: C3 (F (2.075, 31.13) = 5.457, p = 0.0087), T4 (Friedman statistic = 8.925, p = 0.0303). “*” –p < 0.05.

Supplemental Information 54 Postural changes of alpha spectral power (Pα) calculated for C3, C4, T3 and T4 electrodes among female participants during Test 2 (n = 16)

The graphs show confidence intervals with means represented by circle-shaped points, and medians depicted as rhomb-shaped points. Additionally, points and intervals are highlighted by different colors to distinguish between first sitting (oSA) and supine (oHA) positions and second sitting (oSB) and supine (oHB) positions. A nonparametric Friedman test summary for statistically significant results: C4 (Friedman statistic = 18,3, p = 0.0004). “**” –p < 0.01, “***” –p < 0.001.

Supplemental Information 55 Postural changes of alpha spectral power (Pα) calculated for C3, C4, T3 and T4 electrodes among all participants during Test 2 (n = 32)

The graphs show confidence intervals with means represented by circle-shaped points, and medians depicted as rhomb-shaped points. Additionally, points and intervals are highlighted by different colors to distinguish between first sitting (oSA) and supine (oHA) positions and second sitting (oSB) and supine (oHB) positions. A one-way repeated measures ANOVA and a nonparametric Friedman test summaries for statistically significant results: C3 (F (2.52, 78.11) = 5.848, p = 0.0022), C4 (F (2.435, 75.49) = 7.702, p = 0.0004), T3 (F (2.293, 71.08) = 4.265, p = 0.014). “*” –p < 0.05, “**” –p < 0.01.

Supplemental Information 56 Postural changes of alpha spectral power (Pα) calculated for P3, P4, T5 and T6 electrodes among male participants during Test 2 (n = 16)

The graphs show confidence intervals with medians depicted as rhomb-shaped points. Additionally, points and intervals are highlighted by different colors to distinguish between first sitting (oSA) and supine (oHA) positions and second sitting (oSB) and supine (oHB) positions.

Supplemental Information 57 Postural changes of alpha spectral power (Pα) calculated for P3, P4, T5 and T6 electrodes among female participants during Test 2 (n = 17)

The graphs show confidence intervals with means represented by circle-shaped points, and medians depicted as rhomb-shaped points. Additionally, points and intervals are highlighted by different colors to distinguish between first sitting (oSA) and supine (oHA) positions and second sitting (oSB) and supine (oHB) positions. A nonparametric Friedman test summary for statistically significant results: P4 (Friedman statistic = 15.56, p = 0.0014). “**” –p < 0.01.

Supplemental Information 58 Postural changes of alpha spectral power (Pα) calculated for P3, P4, T5 and T6 electrodes among all participants during Test 2 (n = 30)

The graphs show confidence intervals with means represented by circle-shaped points, and medians depicted as rhomb-shaped points. Additionally, points and intervals are highlighted by different colors to distinguish between first sitting (oSA) and supine (oHA) positions and second sitting (oSB) and supine (oHB) positions. A nonparametric Friedman test summary for statistically significant results: P3 (Friedman statistic = 15.35, p = 0.0015), P3 (F (2.324, 65.07) = 4.614, p = 0.0099). “*” –p < 0.05, “**” –p < 0.01.

Supplemental Information 59 Postural changes of alpha spectral power (Pα) calculated for Fz, Cz and Pz electrodes among male participants during Test 2 (n = 16)

The graphs show confidence intervals with medians depicted as rhomb-shaped points. Additionally, points and intervals are highlighted by different colors to distinguish between first sitting (oSA) and supine (oHA) positions and second sitting (oSB) and supine (oHB) positions.

Supplemental Information 60 Postural changes of alpha spectral power (Pα) calculated for Fz, Cz and Pz electrodes among female participants during Test 2 (n = 17)

The graphs show confidence intervals with means represented by circle-shaped points, and medians depicted as rhomb-shaped points. Additionally, points and intervals are highlighted by different colors to distinguish between first sitting (oSA) and supine (oHA) positions and second sitting (oSB) and supine (oHB) positions. A one-way repeated measures ANOVA and a nonparametric Friedman test summaries for statistically significant results: Fz (F (2.085, 33.36) = 3.764, p = 0.032), Cz (Friedman statistic = 13.28, p = 0.0041). “*” –p < 0.05, “**” –p < 0.01.

Supplemental Information 61 Postural changes of alpha spectral power (Pα) calculated for Fz, Cz and Pz electrodes among all participants during Test 2 (n = 33)

The graphs show confidence intervals with means represented by circle-shaped points, and medians depicted as rhomb-shaped points. Additionally, points and intervals are highlighted by different colors to distinguish between first sitting (oSA) and supine (oHA) positions and second sitting (oSB) and supine (oHB) positions. A one-way repeated measures ANOVA and a nonparametric Friedman test summaries for statistically significant results: Fz (F (2.249, 71.96) = 5.319, p = 0.0052), Cz (Friedman statistic = 20.21, p = 0.0002), Pz (Friedman statistic = 14.92, p = 0.0019) “*” –p < 0.05, “**” –p < 0.01, “***” –p < 0.001.

Supplemental Information 62 Sex differences in alpha spectral power (Pα) for F3, F4, C3, C4, P3 and P4 electrodes during first (HA) and last (HB) 2 min of supine position in Test 1 (n= 35)

Black boxplots include values of male participants (m), and red boxplots contain values of female participants (f). Pairs of boxplots were analyzed separately, i.e., HA (m) was compared only to HA (f), and HB (m) was compared only to HB (f). Outliers are shown by black and blue points. A one-way ANOVA and a nonparametric Kruskal–Wallis test summaries for statistically significant results: F3 (F (3, 66) = 6.412, p = 0.0007), F4 (F (3, 66) = 6.543, p = 0.0006), C3 (F (3, 66) = 6.562, p = 0.0006), C4 (Kruskal–Wallis statistic = 12.87, p = 0.0049), P3 (Kruskal–Wallis statistic = 10.07, p = 0.018), P4 (Kruskal–Wallis statistic = 12.45, p = 0.006) “*” –p < 0.05, “**” –p < 0.01.

Supplemental Information 63 Sex differences in alpha spectral power (Pα) for F7, F8, T3, T4, T5 and T6 electrodes during first (HA) and last (HB) 2 min of supine position in Test 1 (n= 35)

Black boxplots include values of male participants (m), and red boxplots contain values of female participants (f). Pairs of boxplots were analyzed separately, i.e., HA (m) was compared only to HA (f), and HB (m) was compared only to HB (f). Outliers are shown by black and blue points. A one-way ANOVA and a nonparametric Kruskal–Wallis test summaries for statistically significant results: F7 (F (3, 66) = 4.9, p = 0.0039), F8 (Kruskal–Wallis statistic = 11.77, p = 0.0082), T3 (Kruskal–Wallis statistic = 9.926, p = 0.0192), T4 (Kruskal–Wallis statistic = 11.25, p = 0.0104), T6 (Kruskal–Wallis statistic = 14.87, p = 0.0019) “*” –p < 0.05, “**” –p < 0.01.

Supplemental Information 64 Sex differences in alpha spectral power (Pα) for F3, F4, F7, F8, C3, C4 and T4 electrodes during sitting positions (oSA and oSB) in Test 2 (n= 33)

Black boxplots include values of male participants (m), and red boxplots contain values of female participants (f). Pairs of boxplots were analyzed separately, i.e., oHA (m) was compared only to oHA (f), and oHB (m) was compared only to oHB (f). A one-way ANOVA and a nonparametric Kruskal–Wallis test summaries for statistically significant results: F3 (F (3, 62) = 3.761, p = 0.0151), F4 (F (3, 62) = 5.345, p = 0.0024), F7 (Kruskal–Wallis statistic = 9.441, p = 0.024), F8 (F (3, 62) = 4.333, p = 0.0078), C3 (F (3, 60) = 2.944, p = 0.04), T4 (Kruskal–Wallis statistic = 10.19, p = 0.017) “*” –p < 0.05, “**” –p < 0.01.

Supplemental Information 65 Sex differences in alpha spectral power (Pα) for F3, F4, C3, C4, P3 and P4 electrodes during supine positions (oHA and oHB) in Test 2 (n= 33)

Black boxplots include values of male participants (m), and red boxplots contain values of female participants (f). Pairs of boxplots were analyzed separately, i.e., oHA (m) was compared only to oHA (f), and oHB (m) was compared only to oHB (f). A one-way ANOVA and a nonparametric Kruskal–Wallis test summaries for statistically significant results: F3 (F (3, 62) = 6.162, p = 0.001), F4 (F (3, 62) = 6.904, p = 0.0004), C3 (F (3, 60) = 4.243, p = 0.0018), C4 (Kruskal–Wallis statistic = 11.64, p = 0.0087), P3 (Kruskal–Wallis statistic = 11.77, p = 0.0082), P4 (F (3, 62) = 4.264, p = 0.0084) “*” –p < 0.05, “**” –p < 0.01.

Supplemental Information 66 Sex differences in alpha spectral power (Pα) for F7, F8, T3, T4, T5 and T6 electrodes during supine positions (oHA and oHB) in Test 2 (n= 33)

Black boxplots include values of male participants (m), and red boxplots contain values of female participants (f). Pairs of boxplots were analyzed separately, i.e., oHA (m) was compared only to oHA (f), and oHB (m) was compared only to oHB (f). A one-way ANOVA and a nonparametric Kruskal–Wallis test summaries for statistically significant results: F7 (F (3, 62) = 5.622, p = 0.0018), F8 (Kruskal–Wallis statistic = 13.76, p = 0.0033), T3 (F (3, 60) = 4.495, p = 0.0065), T4 (F (3, 60) = 4.884, p = 0.0042), T5 (Kruskal–Wallis statistic = 13.27, p = 0.0041). “*” –p < 0.05, “**” –p < 0.01.

Supplemental Information 67 Simple linear regression graphs showing estimated relationship between RWA and Pα in all 4 REG leads and several corresponding EEG electrodes among female participants in Test 1 (n= 8)

Each point of certain color represents estimated relationship between RWA and Pα in the corresponding pair of REG lead and EEG electrode, for example, left FM and F3 (LFM-F3), right OM and C4 (RFM-C4), etc. The lines indicate the general trend of the relationship.

Supplemental Information 68 Simple linear regression graphs showing estimated relationship between RWA and Pα in all 4 REG leads and several corresponding EEG electrodes among all participants in Test 1 (n= 8)

Each point of certain color represents estimated relationship between RWA and Pα in the corresponding pair of REG lead and EEG electrode, for example, left FM and F3 (LFM-F3), right OM and C4 (RFM-C4), etc. The lines indicate the general trend of the relationship.

Supplemental Information 69 Simple linear regression graphs showing estimated relationship between RWA and Pα in all 4 REG leads and several corresponding EEG electrodes among male participants in Test 2 (n= 8)

Each point of certain color represents estimated relationship between RWA and Pα in the corresponding pair of REG lead and EEG electrode, for example, left FM and F3 (LFM-F3), right OM and C4 (RFM-C4), etc. The lines indicate the general trend of the relationship.

Supplemental Information 70 Simple linear regression graphs showing estimated relationship between RWA and Pα in all 4 REG leads and several corresponding EEG electrodes among female participants in Test 2 (n= 8)

Each point of certain color represents estimated relationship between RWA and Pα in the corresponding pair of REG lead and EEG electrode, for example, left FM and F3 (LFM-F3), right OM and C4 (RFM-C4), etc. The lines indicate the general trend of the relationship.

Supplemental Information 71 Simple linear regression graphs showing estimated relationship between RWA and Pα in all 4 REG leads and several corresponding EEG electrodes among all participants in Test 2 (n= 8)

Each point of certain color represents estimated relationship between RWA and Pα in the corresponding pair of REG lead and EEG electrode, for example, left FM and F3 (LFM-F3), right OM and C4 (RFM-C4), etc. The lines indicate the general trend of the relationship

Supplemental Information 72 Dataset of the measured cardiovascular parameters

Supplemental Information 73 REG and EEG dataset of Test 1

Supplemental Information 74 REG and EEG dataset of Test 2

We thank professor Nikolay B. Suvorov, from our department for providing guidance in acquiring electrophysiological data.

List of abbreviations

EEG electroencephalography

REG rheoencephalography

FM fronto-mastoid lead

OM occipito-mastoid lead

RWA reowave amplitude

VO venous output

AT anacrotic time

CT catacrotic time

Pα alpha spectral power

Additional Information and Declarations

Competing Interests

Author Contributions

Human Ethics

Data Availability

The authors declare there are no competing interests.

Arlan F. Sagirov conceived and designed the experiments, performed the experiments, analyzed the data, prepared figures and/or tables, and approved the final draft.

Timofey V. Sergeev performed the experiments, analyzed the data, prepared figures and/or tables, and approved the final draft.

Maria V. Kuropatenko analyzed the data, authored or reviewed drafts of the article, and approved the final draft.

Alexander V. Shabrov analyzed the data, authored or reviewed drafts of the article, and approved the final draft.

Elizaveta A. Agapova performed the experiments, authored or reviewed drafts of the article, and approved the final draft.

Alexey A. Anisimov performed the experiments, authored or reviewed drafts of the article, and approved the final draft.

Alexander Z. Yafarov analyzed the data, prepared figures and/or tables, and approved the final draft.

The following information was supplied relating to ethical approvals (i.e., approving body and any reference numbers):

The Institute of Experimental Medicine gave Ethical approval to carry out the study within its facilities (Ethical Application No. 2/22, dated 06.04.2022)

The following information was supplied regarding data availability:

The raw data is available in the Supplemental Files.

The REG and EEG datasets are available at GitHub and Zenodo:

–https://github.com/Quartzent/EEG-REG-data.git

– Sagirov, A. (2025). EEG-REG data [Data set]. Zenodo. https://doi.org/10.5281/zenodo.17204527

–Sagirov, A. (2024). REG dataset [Data set]. Zenodo. https://doi.org/10.5281/zenodo.14568951

–Sagirov, A. (2024). EEG dataset [Data set]. Zenodo. https://doi.org/10.5281/zenodo.14569015.

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
