# Peer review of "The role of cerebral blood flow volume in cortical inhibition during postural changes"

_PeerJ, doi:10.7717/peerj.20233_

## Round 0.1 · original submission · Major Revisions

Dear Dr. Sagirov,

The reviewers acknowledged the clarity of the writing, the relevance of the research question, and the coherence between the experimental design and the study objectives. Nevertheless, several critical points were raised, particularly concerning the conceptual interpretation of the findings, methodological rigor, and statistical transparency.

Of particular note is the interpretation of alpha-band EEG suppression as a marker of cortical inhibition. While the rationale is biologically plausible, the reviewers emphasized that the functional significance of alpha activity is context-dependent, potentially reflecting either cortical disengagement or functional inhibition. Therefore, a more cautious and theoretically grounded interpretation is warranted, ideally incorporating alternative neurophysiological frameworks or complementary electrophysiological markers.

Further, questions were raised regarding the validity of rheoencephalography (REG) as a surrogate for intracranial hemodynamics. Given REG’s known sensitivity to extracranial blood flow and movement artifacts, the manuscript would benefit from a more robust justification of its use, including detailed descriptions of the procedures employed to minimize signal contamination and cross-modality interference with EEG recordings.

The central hypothesis positing a baroreflex-mediated pathway leading to cortical inhibition—while compelling—remains speculative in the absence of direct physiological measurements (e.g., baroreceptor sensitivity assessments, plasma norepinephrine concentrations, or surrogate markers such as pupilometry or heart rate variability). The reviewers strongly recommended a more conservative framing of this claim and suggested the integration of contemporary neurovisceral models to support the interpretation of EEG-REG associations.

In terms of statistical analysis, concerns were raised regarding the use of Pearson correlations in the context of non-Gaussian distributions. Justification for this choice should be explicitly provided, or alternative non-parametric methods should be employed. Moreover, an a priori power analysis should be conducted and reported, and clarity is required as to whether alpha power refers to absolute or relative values. The absence of instructions provided to participants during each experimental phase, as well as the lack of state anxiety assessments, were also identified as methodological limitations.

Further points include the need to report results for the entire sample in addition to the sex-stratified analyses. The observed sex differences would be more robustly interpreted if hormonal status (e.g., menstrual cycle phase or estrogen levels) were accounted for. In the absence of such data, the limitations of the current stratification approach must be explicitly acknowledged.

Finally, the methodological decision to pause EEG and REG data acquisition during tilt-table oscillations prevents the evaluation of real-time neurovascular responses, thereby limiting the interpretability of dynamic postural effects. The rationale for this choice should be clearly articulated and its implications addressed in the discussion.

In view of these considerations, we invite you to revise your manuscript in accordance with the reviewers’ recommendations. A comprehensive and structured point-by-point response to all comments should accompany the revised submission, clearly indicating how each issue has been addressed and referencing specific sections of the updated manuscript.

We remain appreciative of your contribution and encourage you to carefully incorporate the necessary refinements so that your work may reach its full potential in terms of scientific rigor and impact.

Sincerely,
Dr. Gustavo Rodrigues Pedrino
Academic Editor
PeerJ

·

Basic reporting

The article titled “The role of cerebral blood flow volume in cortical inhibition during postural changes” investigates how changes in body posture, particularly moving to a supine position, affect cerebral blood flow and cortical activity, revealing that increased cerebral venous volume is associated with reduced alpha wave power—suggesting cortical inhibition primarily driven by baroreflex mechanisms rather than blood flow volume alone, with notable sex-based differences and limited effects of dynamic tilting.

You interpret reductions in alpha spectral power as indicative of cortical inhibition. How do you account for the dual interpretation of alpha power as both a marker of cortical disengagement and functional inhibition depending on context (e.g., sensory gating vs. task disengagement)? Could alternative electrophysiological markers offer a more specific index of inhibition?
What specific model of neurovascular coupling or central-autonomic interaction underlies your hypothesis? Would integrating recent computational models of autonomic control (e.g., from neurovisceral integration frameworks) alter the interpretation of your EEG-REG correlations?

Experimental design

REG is known to be sensitive to extracranial blood flow and motion artifacts. How did you validate the accuracy of REG measurements for intracranial flow, and what steps were taken to isolate cerebral signals from potential myogenic or systemic noise sources?
Since REG electrodes were placed near frontal and occipital EEG sites, could current injection or bioimpedance fluctuations interfere with adjacent EEG signals? Was any shielding or correction applied to minimize cross-modality artifacts?

Validity of the findings

Alpha suppression is context-dependent and not a definitive marker of inhibition across all brain states. Without cognitive or behavioral tasks to verify functional inhibition, the claim of “cortical inhibition” remains somewhat speculative.
The claim that baroreflex activity is the primary driver of cortical inhibition is inferred indirectly through posture-induced cardiovascular changes and cannot be confirmed without direct baroreceptor sensitivity testing (e.g., neck suction, phenylephrine method).

Additional comments

This study would benefit from:

Reframing conclusions more conservatively.

Strengthening theoretical models of alpha activity and baroreflex interactions.

·

Basic reporting

This study associates brain activity and blood supply in different body positions in males and females.
The introduction is well written and the references are updated.

It is quite important to clarify whether the authors refer to alpha absolute of relative power, otherwise some statements may be wrong

The findings are in line with earlier results

Experimental design

The study is in line with the aims of the journal.
The study protocol is correct

Since non-parametric analysis was performed, the authors should indicate the lack of comparisons between females and males as a limitation of the study. Also, it could be interesting to include the analysis of the entire sample independently of sex

Why Pearson instead of Spearman correlations were studied for non-gaussian distributions requiring non parametric analysis?

A priori power analysis could be useful

please report the instruction given to participants for any phase of the experiments
Also, a preliminary assessment of the participants state anxiety would have been useful

Validity of the findings

Please present also the entire sample analysis, provide a priori power analysis, clarify why Pearson rather than Spearman coefficient was used, report the instruction given to participants for any phase of the experiments

Reviewer 3 ·

Basic reporting

n

Experimental design

n

Validity of the findings

n

Additional comments

This study investigates how changes in body posture influence cerebral hemodynamics and cortical activity, focusing on alpha wave suppression as a marker of cortical inhibition. Forty healthy participants underwent EEG and rheoencephalography (REG) recordings during two postural tests: static transitions between sitting and supine positions, and dynamic tilting on a +10°/–10° tilt table. Supine posture significantly increased REG parameters, rheographic wave amplitude, venous outflow, and catacrotic time, especially in males, and was associated with decreased alpha spectral power in EEG, suggesting cortical inhibition. A strong negative correlation between cerebral blood flow volume and alpha power was observed in males, particularly during static postural changes. Females showed less pronounced changes, possibly due to baseline cerebrovascular differences. Dynamic tilting had minimal effect on REG and inconsistent effects on alpha activity. These findings suggest that baroreflex activation, rather than cerebral blood volume alone, drives cortical inhibition, and highlight EEG-REG as a potential tool for assessing baroreflex-related dysfunction. This is an interesting submission.

1. The authors propose that baroreflex activation leads to reduced central norepinephrine output and, in turn, cortical inhibition. This “baroreflex →NE decrease → cortical inhibition” pathway is biologically plausible - after all, lying down facilitates sleep—but the study stops short of directly measuring norepinephrine levels or locus coeruleus activity. Although challenging in humans, techniques heart rate, pupil diameter or dynamic oxygen tension offer more supporting evidence.
2. In Method “Simultaneous EEG and REG recordings were conducted in each position, except during the passive oscillatory movements on the tilt table.” Pausing data acquisition during the ±10° oscillations means we only see pre and post tilt values (oHA vs. oHB) and miss the immediate hemodynamic and electrophysiological responses. Also, a ±10° tilt is relatively mild - would a larger angle (e.g. ±30° or ±45°) yield more pronounced effects?
3. All subjects were young, healthy adults (mean age ≈ 21). Including older individuals or patients with autonomic or neurodegenerative disorders could greatly enhance the study’s clinical relevance.
4. The paper reports clear sex differences in RWA, VO, and α power, but does not account for menstrual cycle phase or measure estrogen levels. Controlling for these variables would strengthen the interpretation of gender effects.

---

## Round 0.2 · accepted · Accept

The authors have thoroughly addressed all of the reviewers’ comments and suggestions. The revisions have improved the clarity and quality of the manuscript. I therefore consider the manuscript ready for publication.

·

Basic reporting

The authors responded adequately to the reviewers’ questions and improved the article.

Experimental design

-

Validity of the findings

-

·

Basic reporting

All my points have been addressed successfully
This section accords with the journal's criteria

Experimental design

This section has been modified according to the journals' criteria.

Validity of the findings

The journal's guidelines have been followed